# Efficacy of Lesion Specific Portals in Endoscopic Treatment of Calcaneal Bone Cyst: A Case Report and Literature Review

**DOI:** 10.3390/medicina57020111

**Published:** 2021-01-26

**Authors:** Young Yi, Jeong Seok Lee, Jahyung Kim, So Young Jin, Sung Hun Won, Jaeho Cho, Dong-Il Chun

**Affiliations:** 1Department of Orthopaedic Surgery, Seoul Foot and Ankle Center, Inje University Seoul Paik Hospital, 85, 2-ga, Jeo-dong, Jung-gu, Seoul 04551, Korea; 20vvin@naver.com; 2Department of Orthopaedic Surgery, Soonchunhyang University Seoul Hospital, 59, Daesagwan-ro, Yongsan-gu, Seoul 04401, Korea; 124856@schmc.ac.kr (J.S.L.); hpsyndrome@naver.com (J.K.); orthowon@gmail.com (S.H.W.); 3Department of Pathology, Soonchunhyang University Seoul Hospital, 59, Daesagwan-ro, Yongsan-gu, Seoul 04401, Korea; jin0924@schmc.ac.kr; 4Department of Orthopaedic Surgery, Chuncheon Sacred Heart Hospital, Hallym University, Chuncheon 24253, Korea; hohotoy@nate.com

**Keywords:** calcaneal bone cysts, benign bone tumors, endoscopic curettage, portal

## Abstract

*Background:* Calcaneal bone cysts rarely occur and most of them are known to be benign. Among them, simple bone cysts (SBCs) third most commonly occur in the calcaneus and of the many surgical treatment options, endoscopic curettage is recently gaining popularity among surgeons due to its advantages of minimal invasiveness and optimal visualization. As for portal placement for endoscopy, two lateral portals are considered a standard technique, but no rationale has been established for SBCs with abnormal geometry. This case report suggests an SBC with secondary aneurysmal change located outside the Ward’s triangle, as well as an appropriate endoscopic approach. *Case Presentation:* An 18-year-old male high school student presented with a main complaint of pain at the hind foot level for the past one year, without significant improvement from conservative treatment. An endoscopic curettage through the lesion specific two posterior portals and bone graft using allogeneic cancellous bone were performed. SBC with a secondary aneurysmal bone cyst was diagnosed on pathology. At a one-year follow-up, the patient was painless and had returned to his regular activities. Physical and radiographic examinations revealed that the lesion was completely healed without any evidence of recurrence. *Conclusion:* For calcaneal bone cysts located at the posterior aspect of the calcaneus, eccentrically medial and abnormally long anterior-posteriorly, we suggest an endoscopic procedure using lesion specific portals such as two posterior portals.

## 1. Background

The calcaneus is a rare location for the development of primary tumors and most of these lesions are represented to be benign [1]. Of the many kinds of benign bone cysts occurring in the calcaneus, simple bone cysts (SBCs) are known to be the most common type, followed by aneurysmal bone cysts or osteochondromas. These multi-loculated, fluid-filled benign lesions enlarge over time, resulting in thinning of the bone cortex. In terms of epidemiology, about 85% of SBCs occur in children and adolescents [2].

Although the symptoms include pain in the heel, most patients have no pain unless in cases of rarely occurring pathologic fracture or undisplaced stress fracture, and some are diagnosed incidentally by plain radiographs [2]. Furthermore, most SBCs eventually heal by the time of physeal closure, and patients older than 10 years heal at a higher rate compared to younger patients regardless of the treatment method [3]. Therefore, the treatment of this benign bone tumor remains controversial. Surgical options should be considered for patients with persistent pain, obvious or suspected pathologic fractures, or those whose cysts continue to enlarge [4].

Of the many surgical treatment options, endoscopic curettage has recently gained popularity among surgeons due to its advantages of minimal invasiveness and optimal visualization [5]. Such a procedure is possible because SBCs are benign and can be treated effectively through intralesional curettage and cyst excision with bone graft [6]. Concerning the portals for approaching the cyst, most surgeons prefer two laterally located portals since most SBCs are located at the base of the lateral calcaneus neck within an area known as Ward’s triangle and are easily accessible from the lateral side by penetrating the thinned lateral wall [7,8,9,10]. However, portals suitable for abnormally medially located calcaneal cysts, which present with thick lateral walls, have not been clearly established, probably due to their rare nature.

Here, we report a rare case of a patient who presented with a calcaneal bone cyst located outside the Ward’s triangle and was successfully treated using the endoscope through the portals that made the lesion specific approach possible.

## 2. Case Presentation

This case report was approved by the Institutional Review Board of Soonchunhyang University Seoul Hospital (Institutional Review Board number: SCHUH 2020-05-001; Date of Approval: 7 May 2020) and the patient gave written informed consent for publication of this report and the accompanying images.

### 2.1. Preoperative Evaluation

An 18-year-old male high school student presented with a main complaint of pain at the hindfoot level for the past one year, without significant improvement from conservative treatment. The patient’s pain intensity was mild-to-moderate, which was aggravated by activity. He had no previous trauma, family history, or other pathological conditions on the affected foot. On physical examination, localized pain on palpation was detected on the medial side of the hindfoot. The sensory and motor exams revealed no deficits and the lab tests were within the normal range.

Plain radiographs (lateral and axial views of the calcaneus) revealed a 4.5 × 2.5 × 4.6 cm-sized posteromedially eccentrically located, well-marginated lytic lesion, with thinning of the posterior and medial cortex in the left calcaneus. No penetration or periosteal reaction was found (Figure 1). Computed tomography (CT) suggested internal calcification of the lesion without any sign of stress fracture (Figure 2). Magnetic resonance imaging (MRI) confirmed a multi-lobulated cystic structure with intermediate T1 and heterogeneously high T2 signal intensity. The cyst was peripherally enhanced and fluid-fluid level formation was also found within the structure (Figure 3). The Foot and Ankle Outcome Score (FAOS) was 73 points [11]. Based on the patient’s age and the radiographic examination, the cyst was expected to be calcaneal chondroblastoma with secondary aneurysmal change and endoscopic curettage with bone graft, usually performed treatment modality, was planned [12].

### 2.2. Surgical Procedure

Surgery was performed under lumbar spinal anesthesia with the patient in the prone position and with the application of a tourniquet at the thigh level.

First, the location of the calcaneal bone cyst was confirmed under an image intensifier and a spinal needle was inserted at the medial aspect of the bone cyst and aspiration of the intra-cystic fluid was performed (Figure 4). The aspirated fluid was reddish, clear, and plasma-like.

Under an image intensifier, one guidewire was inserted from the most thinned wall of the posteromedial aspect of the cyst toward the center. The other guidewire was inserted through the posterolateral aspect of the cyst, forming a right angle with the previously inserted guidewire (Figure 5). Using a 5.0 mm diameter cannulated drill, the posteromedial and posterolateral portals were penetrated following the inserted guidewires and an endoscope was inserted (Figure 6).

In conjunction with a thorough inspection of the cystic structure using both 30° and 70° endoscope, a biopsy sample was obtained and sent to the pathology department for histologic examination. Using a suction shaver and small curette, resection of the inner bony septum was performed to reduce the endoscopic blind area. Under endoscopic visualization, the fibrous inner surface of the cyst was circumferentially resected and debrided with a curette and abrader with special care not to provoke an iatrogenic fracture of the thinned cortical wall (Figure 7). In an effort to perform a complete curettage of the cyst, repetitive debridement was performed by swapping the viewing and working portals. After massive irrigation, the allogenic cancellous bones were inserted inside the cyst through the portals and were impacted (Figure 8). 

### 2.3. Postoperative Progression

The histologic sample showed a cystic wall containing ossification along the fibrous wall. Within the wall, there also were fibrous exudate, hemosiderin-pigmented macrophages, and focal giant cell reaction due to Aneurysmal bone cystic changes. The final diagnosis was reported as an SBC with a secondary aneurysmal bone cyst (Figure 9).

The patient was advised to perform non-weight-bearing walking on the affected limb for six weeks. After additional four weeks of partial weight-bearing, he was allowed to fully bear weight and return to his regular activity.

At one year of follow-up, the patient was painless and the FAOS improved to 100 points. Physical and radiographic examinations revealed that the lesion was completely healed without any evidence of recurrence (Figure 10).

## 3. Discussion

Since first introduced by Bonnel et al., a minimally invasive endoscopic procedure for calcaneal cyst is gaining preference as an alternative to open surgical procedures [13]. As a result, a variety of papers describing calcaneal bone cysts successfully treated with endoscopic curettage have been reported in the literature (Table 1) [5,13,14,15,16,17,18,19,20,21,22]. Endoscopic curettage is advantageous in many ways. First, an accurate evaluation of the extent of the lesion is possible owing to the direct visualization of the cyst wall and contents. Second, the risk of iatrogenic fracture can be reduced because the endoscope enables a surgeon to assess the adequacy of the curettage, thus avoiding excessive and needless curettage. Third, the recurrence rate can possibly be decreased with the ability to completely evacuate the lesion. Lastly, the procedure can be performed with minimal blood loss and small postoperative scars [23].

Different from ordinary arthroscopy, which directly approaches the joint by passing through soft tissue, the cortical wall of the calcaneus needs to be fenestrated in order to visualize the internal structure of the bone cyst. This is why some authors differentiate this procedure with the term calcaneal “ossoscopy” instead of “endoscopy” [10]. Therefore, understanding the accurate geometry and internal structure of the calcaneus bone cyst needs to be preceded due to the risk of iatrogenic wall fracture through inappropriate portal fenestration. 

By retrospectively analyzing 47 patients with SBCs, Pagoda et al. reported that calcaneal cysts appeared as well-circumscribed areas of oval lucency within the anterior portion of the calcaneus below the calcaneal sulcus and the posterior articular surface of the talus [24]. Another study stated that two-thirds of the benign calcaneal lesion cysts were restricted to the inferior anterior part of the bone [25]. Also, cyst laterality was evaluated on the coronal view of the T2-weighted MRI. In an evaluation of 24 cases, the cysts were centrally located in 21 cases (87.5%), medially in two cases (8.3%), and laterally in one (4.2%) [7]. In summary, SBCs are usually located inside the calcaneus centrally, anteriorly, and inferiorly, within the Ward’s triangle, making the affected side wall thin. Such anatomical properties of SBCs make it easier for most surgeons to approach the cyst using lateral portals [5,8,9]. Lateral portals are also preferred over medial portals to minimize the possible iatrogenic neurovascular injury [26].

However, the abnormal geometry of the bone cyst in this case made us hesitant to choose the widely used two lateral portals. In the present case, the cyst was located far medially, presenting a thinned medial wall and a thickened lateral wall. We predicted that the length of the portal tunnel would become too long, which would eventually make the endoscope fixed within the cortical bone structure. Such a situation would force us to make the cortical window diameter larger to allow free movement of the endoscope, increasing the risk of iatrogenic fracture [27,28].

Another factor to consider was that the anterior-posterior diameter of the cyst in our case was longer compared with cysts presented in other studies [23,29,30,31], making the area needing debridement wider. We thought that if we performed debridement through lateral portals, where the range of motion for the suction shaver would be limited due to the thick wall, sufficient debridement of the unusually broad area of the medial and lateral cystic wall would not be possible (Figure 11A). Due to these limitations, we had to find other portals to access the cyst.

As an alternative, we chose two posterior portals inserted through the most rounded side of the cyst. The posterior portals were advantageous in our case because of the posteriorly located geometry, making it easier to fenestrate through the thinned part of the wall. As an extension, the relatively thin posterior wall of the cyst enabled us to move the suction shaver and curette within the cyst without any disturbance (Figure 11B). Through these lesion specific portals, we were able to sufficiently debride the calcaneal wall. 

In the literature, the recurrence rate of secondary aneurysmal bone cyst is reported to be high [32]. In addition, secondary aneurysmal bone cysts usually occur secondary to giant cell tumor, osteoblastoma, or chondroblastoma and not much has been known about the recurrence after therapeutic approach in SBC with aneurysmal bone cyst [29]. For this reason, one year follow-up period would have been relatively short for closely monitoring the recurrence of the lesion, although gratifying radiographic outcome and sufficient patient satisfaction without pain or limitation in daily activity were achieved. 

## 4. Conclusions

We described a case of a calcaneal bone cyst with secondary aneurysmal bone cyst within abnormal geometry that was successfully treated with an endoscopic curettage and bone graft. Such success may be attributed to the use of novel, lesion-specific endoscopic portals in consideration of specific issues like excessive cortical fenestration and limited working portal mobility. Therefore, with the use of lesion-specific portals instead of conventional two lateral portals, we propose that endoscopic treatment toward calcaneal bone cyst can be successfully achieved regardless of the cyst geometry.

## Figures and Tables

**Figure 1 medicina-57-00111-f001:**
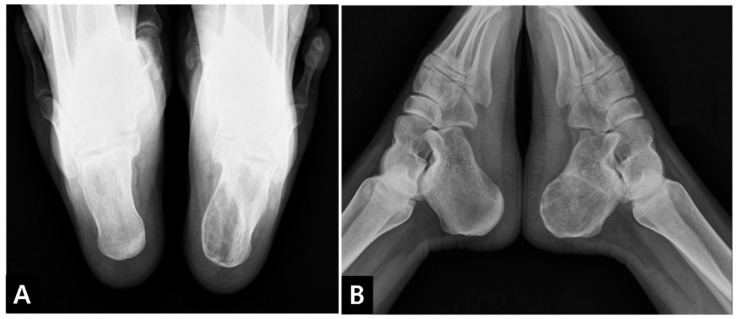
Preoperative radiographs of the calcaneus. (**A**) axial and (**B**) lateral view.

**Figure 2 medicina-57-00111-f002:**
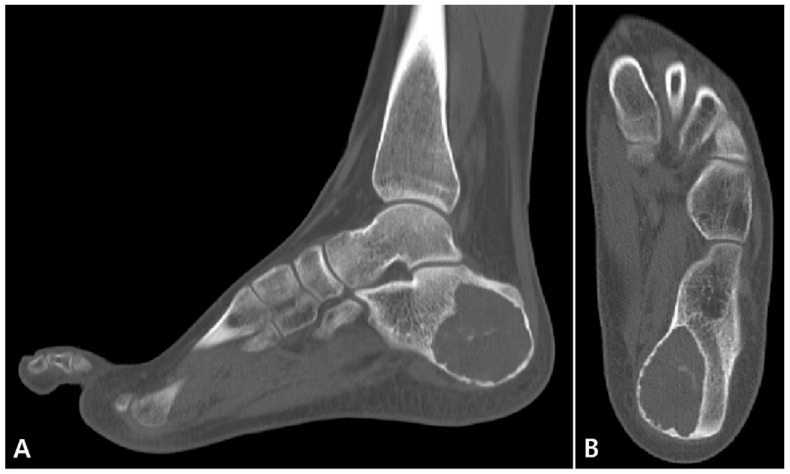
Preoperative computed tomography (CT) scans of the left foot (**A**) sagittal and (**B**) axial views show lytic mass lesion, 4.6 × 2.4 × 4.4 cm in dimension, localized postero-medially, leading to the cortical bone thinning, containing calcified lesions without signs of stress fracture.

**Figure 3 medicina-57-00111-f003:**
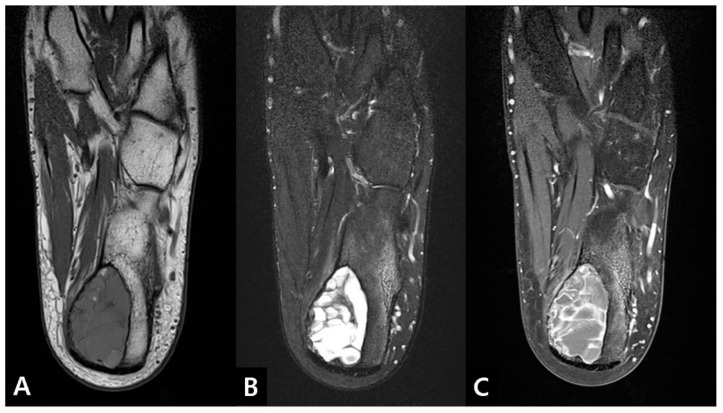
Preoperative magnetic resonance imaging (MRI) scans axial view shows a cystic mass lesion in the posteromedial part of the calcaneus medullary cavity. (**A**) T1W intermediate, (**B**) T2W heterogeneously high signal intensity multi-lobulated cystic structure. (**C**) Contrast enhanced image shows a peripherally enhanced mass with fluid-fluid level formation.

**Figure 4 medicina-57-00111-f004:**
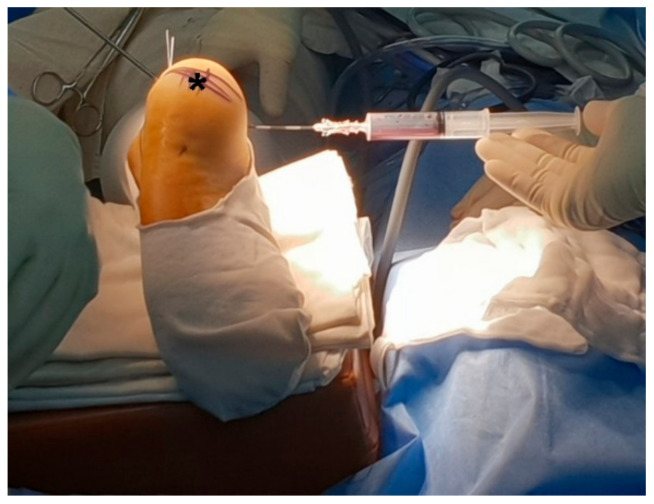
Intraoperative aspiration of the cyst. The patient is in a prone position. The black asterisk (*) marks the calcaneus.

**Figure 5 medicina-57-00111-f005:**
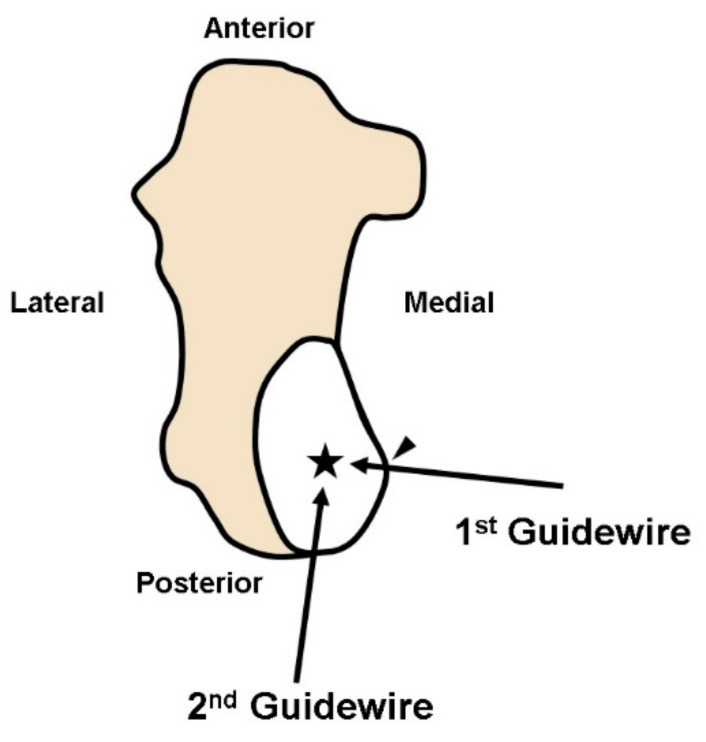
Portal placement. The first guide wire inserted from the most thinned wall of the posteromedial aspect of the cyst (arrowhead) toward the center (asterisk). The second guidewire inserted through posterolateral aspect of the cyst, forming a right angle with first guidewire.

**Figure 6 medicina-57-00111-f006:**
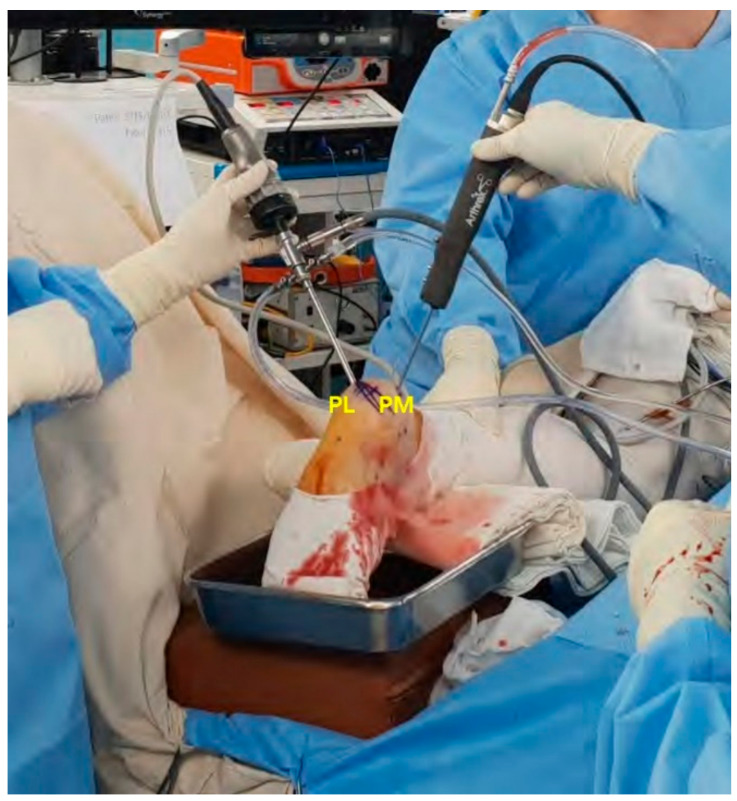
Intraoperative image showing the posterolateral viewing portal (PL) and the posteromedial working portal (PM).

**Figure 7 medicina-57-00111-f007:**
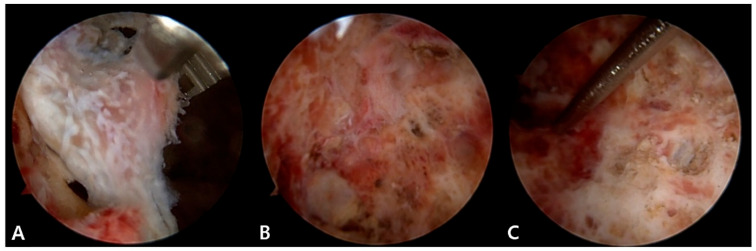
Endoscopic view showing (**A**) the inner septum in the cyst and (**B**) the cavity after resection of the bony septum. (**C**) After curettage of the inner surface of the calcaneal cyst.

**Figure 8 medicina-57-00111-f008:**
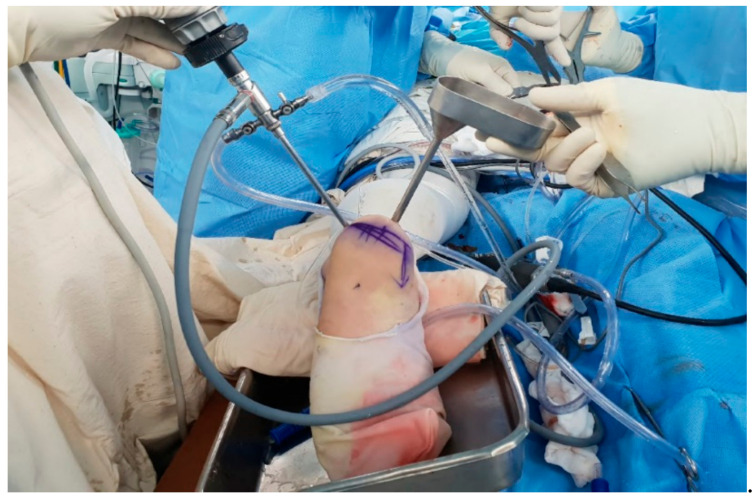
Impaction of the allogenic cancellous bone chips through the portals.

**Figure 9 medicina-57-00111-f009:**
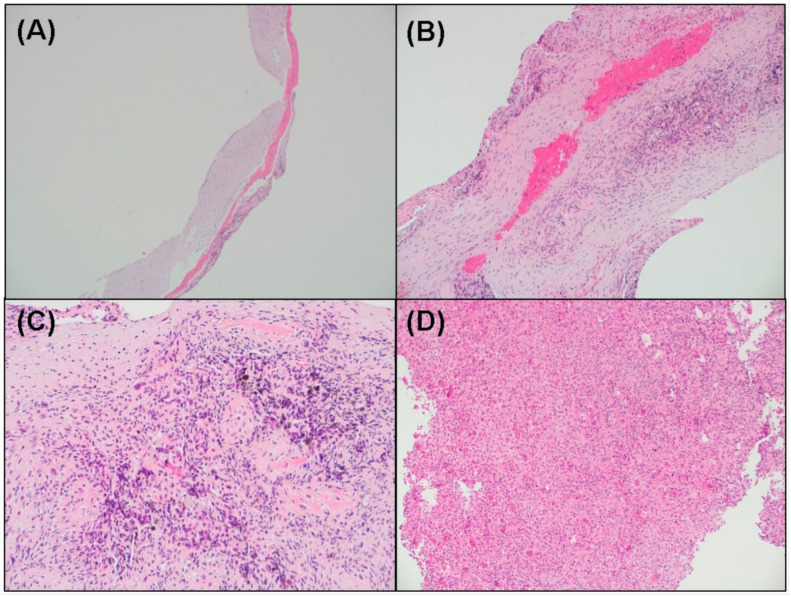
Histologic features of the obtained specimen. (**A**): Cystic wall containing ossification along the fibrous wall (Hematoxylin-eosin stained (H-E); magnification, ×40). (**B**) Fibrous exudate in the wall (H-E; magnification, ×40). (**C**) Hemosiderin pigment in the wall (H-E; magnification, ×40). (**D**) Focal giant cell reaction due to Aneurysmal bone cystic change (H-E; magnification, ×100).

**Figure 10 medicina-57-00111-f010:**
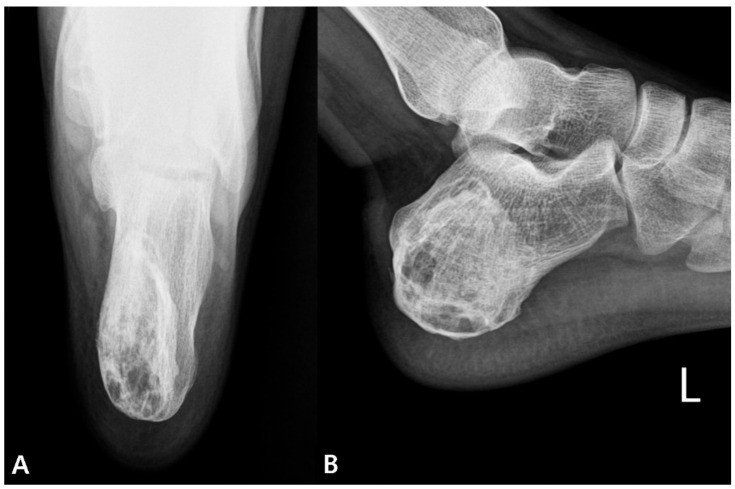
Postoperative one year (**A**) axial and (**B**) lateral view of the calcaneus.

**Figure 11 medicina-57-00111-f011:**
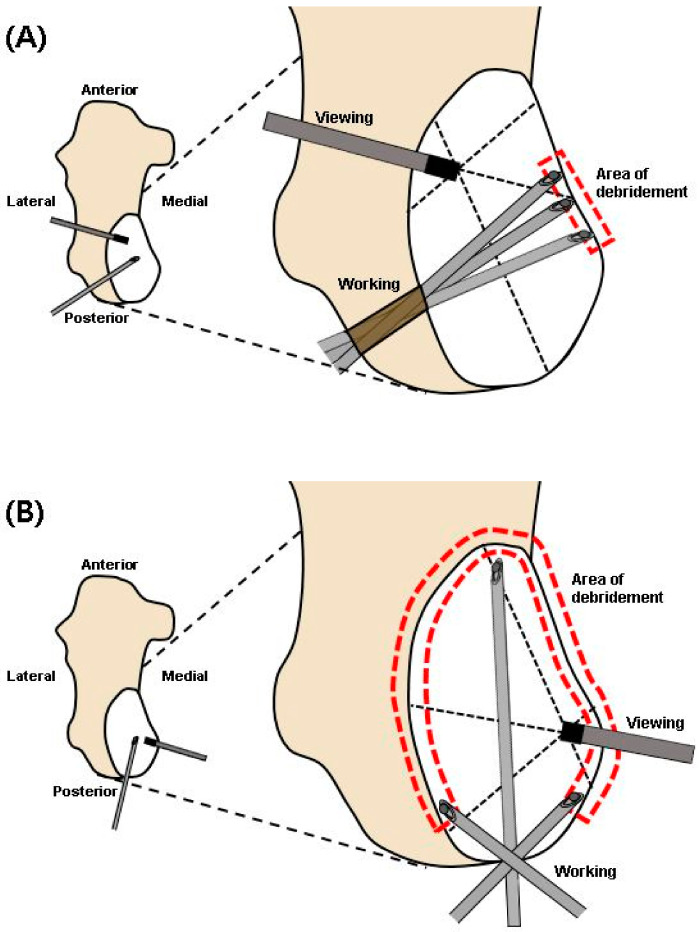
Comparison of the accessible area for debridement (dotted area) upon the portal placement. Working on the lateral portal (**A**) provides limited motion for the suction shaver because of the thick cyst wall. Accessed from the posterior portal (**B**), the suction shaver is more mobile because of the thin cyst wall, allowing a larger area for debridement.

**Table 1 medicina-57-00111-t001:** Review of the literature on the endoscopic treatment for calcaneal bone cysts.

Authors	#	Type of Cyst	Portal Placement	Graft Material after Curettage	Recurrence
Mainard et al. [5] (2006)	1	SBC	One lateral	Calcium phosphate cement	No
Innami et al. [8] (2011)	10	SBC	Two lateral	Calcium phosphate cement	No
Stoica et al. [9] (2017)	1	SBC	Two lateral	Allograft	No
Bonnel et al. [13] (1999)	1	SBC	Two lateral	Autograft	No
Aiba et al. [14] (2018)	6	SBC	One medial, One lateral	None	No
Choi et al. [15] (2014)	22	SBC (9), Fibrous dysplasia (6) Enchondroma (5) Intraosseous lipoma (1) Brodie abscess (1)	Two lateral	Autograft or Allograft	Yes (One case)
Jung et al. [16] (2008)	1	SBC	Two lateral	Autograft	No
Koo et al. [17] (2001)	3	SBC	Two lateral	Autograft or Allograft	No
Nishimura et al. [18] (2016)	8	SBC	Two lateral	Calcium phosphate cement	No
Yildirim al. [19] (2011)	13	SBC	Two lateral	Allograft	No
Toepfer et al. [22] (2016)	6	SBC (6) IOL (4)	Two lateral	Allograft	No

# = number of cases, SBC = simple bone cyst, ABC = aneurysmal bone cyst, IOL = intra-osseous lipoma, DBM = demineralized bone matrix, PMMA = polymethylmethacrylate.

## Data Availability

Data sharing is not applicable to this article as no datasets were generated or analyzed during the current study.

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
