# Peer review of "Efficacy of Lesion Specific Portals in Endoscopic Treatment of Calcaneal Bone Cyst: A Case Report and Literature Review"

_medicina, 2021, doi:10.3390/medicina57020111_

Round 1

Reviewer 1 Report

This is a good case report that clearly suggests that the accuracy of endoscopic technique can be further improved by adding the author's lesion-specific portal procedure to the technique presented in the previous reports.
Please check the following points.

1. Why do you think the results were good in previous studies that did not use the lesion specific portal? Doesn't the degree of marginal debridement affect clinical and radiological outcomes?

2. In the axial view of Figure 11, the sclerotic margins on lateral side of the lesion are specifically observed even after surgery. Is not the endoscopic debridement insufficient even with the lesion specific portal?

3. Line #35-36 “the” → them? Please clarify.

4. Line #37 “calcanus” → calcaneus?

5. Line #51-52 “Such a procedure is possible because SBCs are benign and treatment does not require a margin.”
Please clearly describe the author's intention for the term “margin”.

6. Line #67-75 “An 18-year-old male high school student presented with a main complaint of pain at the hindfoot level for the past one year, without significant improvement from conservative treatment. The patient’s pain was of mild-to-moderate intensity, which was partially relieved by anti-inflammatory medication and rest and aggravated by activity. There was no personal or family history of trauma or other pathological conditions at the level of the affected foot. The general clinical exam showed no pathological characteristics. On physical examination, localized pain on palpation was identified on the medial side of the hindfoot. The sensory and motor exams revealed no deficits and the lab tests were within the normal range.”

The arrangement of some words in the text is similar to the arrangement in the following literature. This can lead to unnecessary misunderstandings.

Rom J Morphol Embryol. 2017;58(2):689-693. Unicameral bone cyst of the calcaneus - minimally invasive endoscopic surgical treatment. Case report
“with the main complaint of pain at hindfoot level during the last six months, without significant improvement with conservative treatment. Pain was of mild to moderate intensity; it was partially relieved by anti-inflammatory medication and rest and aggravated by activity. There was no personal or family history of trauma or other pathological conditions at the level of the affected foot. The general clinical exam showed no pathological characteristics. On physical examination, localized pain on palpation was identified on the lateral side of the hindfoot. Lab tests were within the normal range.”

7. Line #143. The Figure 10 is not cited in the manuscript. Please add it to the appropriate position.

8. Line #148-149. “Later he was mobilized with partial weight-bearing walking for an additional four weeks, followed by full weight-bearing on the affected limb.”
Please correct it with the native expression.

9. Line #161-166 “First, the direct visualization of the cyst wall and contents permits an accurate assessment of the extent of the lesion. Second, the endoscope permits an accurate assessment of the adequacy of the curettage, thus avoiding the need to perform multiple, blind, and aggressive passes with a curette, which can increase the risk of iatrogenic fracture. Third, the ability to completely evacuate the lesion should logically reduce the rate of recurrence. Lastly, it has advantages of a small incision, minimal blood loss, and limited dissection.”

The arrangement of some words in the text is similar to the arrangement of the following literature.

J Foot Ankle Surg. Jan-Feb 2010;49(1):93-7. doi: 10.1053/j.jfas.2009.08.005. Treatment of a unicameral bone cyst of calcaneus with endoscopic curettage and percutaneous filling with corticocancellous allograft
“First, direct visualization of the cyst wall and contents permits accurate assessment of the extent of the lesion. Second, the endoscope permits accurate assessment of the adequacy of the curettage, thus avoiding the need to perform multiple, blind, and aggressive passes with a curette, which can increase the risk of violation of the cortical shell and may prolong the procedure. Third, the ability to completely evacuate the lesion should logically reduce the rate of recurrence. In the patients depicted in this technique, the authors used endoscopic curettage because it provided optimal visualization with the advantage of a small incision, minimal blood loss, and a limited dissection.”

10. Thank you for your hard work again.

Author Response

  1. Why do you think the results were good in previous studies that did not use the lesion specific portal? Doesn't the degree of marginal debridement affect clinical and radiological outcomes?

-> Thank you for your detailed comment. As we described on Table 1, good results have been reported in many of the previous studies using standard two lateral portals. We found out that most of the cysts in these studies were within the “Ward’s triangle”, which can typically be approached and treated appropriately with the standard portals. However, the cyst in our case was located outside the Ward’s triangle, which made it difficult for us to approach through the conventional lateral portals. Because of such abnormal geometry of the cyst, we had to come up with lesion-specific portals. We wanted to emphasize in this study was that with the use of lesion-specific portals, successful endoscopic treatment can be achieved regardless of the cyst geometry.

 In the literature, the main components of treatment of SBC include adequate decompression of the intraosseous pressure, irrigation of the cyst to decrease the bone-destroying enzyme, eradication of cyst membrane activity, and stimulation of bone healing process (Hou, H, Treatment of Unicameral Bone Cyst, J Bone Joint Surg Am. 2010;92:855-62). As a result, intralesional curettage and cyst excision with bone graft is known to be optimal modality of treatment (Kadhim M, Treatment of unicameral bone cyst: systematic review and meta analysis. J Child Orthop. 2014;8:171– 91). We agree with you that degree of marginal debridement may affect clinical and radiological outcomes because it may contribute to eradication of cyst membrane activity and stimulation of bone healing process. Given the chance, it would be helpful to perform a study to focus on the correlation between the degree of marginal debridement and postoperative outcomes.

  1. In the axial view of Figure 11, the sclerotic margins on lateral side of the lesion are specifically observed even after surgery. Is not the endoscopic debridement insufficient even with the lesion specific portal?

-> Thank you for your comment. We agree with you that endoscopic debridement may not sufficient surgical option to get rid of the tumor completely, as seen in our postoperative radiograph. However, we want to carefully insist that endoscopic debridement is a permissible treatment modality in SBC because the tumor is known to be benign and does not require marginal or radical resection. Besides, we performed repetitive debridement by swapping the viewing and working portals to get rid of the tumor and to eradicate the cyst membrane as much as possible. In addition, we used both 30’ and 70’ scopes to minimize the possible blind spots within the cyst that can be made near the portals. As a result, intraoperative endoscopic findings indicated that sufficient debridement was made. Indeed, in our opinion, we cannot exclude the possibility that sclerotic margins seen on the radiograph may not be a typical finding suggestive of insufficient eradication of the cyst membrane.

In the following case, we will try to get rid of the tumor meticulously, focusing on the sclerotic margins of the cyst, by confirming with an intraoperative C-arm intensifier. We hope you will understand our opinion. Thank you.

  1. Line #35-36 “the” → them? Please clarify.

-> Thank you for your comment. We revised the manuscript accordingly (Line 35-36).

  1. Line #37 “calcanus” → calcaneus?

-> Thank you and we revised the manuscript accordingly (Line 37).

  1. Line #51-52 “Such a procedure is possible because SBCs are benign and treatment does not require a margin.”
    Please clearly describe the author's intention for the term “margin”.

-> Thank you and we clarified the manuscript (Line 51-52).

  1. Line #67-75 “An 18-year-old male high school student presented with a main complaint of pain at the hindfoot level for the past one year, without significant improvement from conservative treatment. The patient’s pain was of mild-to-moderate intensity, which was partially relieved by anti-inflammatory medication and rest and aggravated by activity. There was no personal or family history of trauma or other pathological conditions at the level of the affected foot. The general clinical exam showed no pathological characteristics. On physical examination, localized pain on palpation was identified on the medial side of the hindfoot. The sensory and motor exams revealed no deficits and the lab tests were within the normal range.”

The arrangement of some words in the text is similar to the arrangement in the following literature. This can lead to unnecessary misunderstandings.

Rom J Morphol Embryol. 2017;58(2):689-693. Unicameral bone cyst of the calcaneus - minimally invasive endoscopic surgical treatment. Case report
“with the main complaint of pain at hindfoot level during the last six months, without significant improvement with conservative treatment. Pain was of mild to moderate intensity; it was partially relieved by anti-inflammatory medication and rest and aggravated by activity. There was no personal or family history of trauma or other pathological conditions at the level of the affected foot. The general clinical exam showed no pathological characteristics. On physical examination, localized pain on palpation was identified on the lateral side of the hindfoot. Lab tests were within the normal range.”

-> Thank you and we revised the paragraph into our own words (Line 68-74).

  1. Line #143. The Figure 10 is not cited in the manuscript. Please add it to the appropriate position.

-> Thank you for your comment. We rearranged the incorrectly written numbers of the figures (Line 140, 151).

  1. Line #148-149. “Later he was mobilized with partial weight-bearing walking for an additional four weeks, followed by full weight-bearing on the affected limb.”
    Please correct it with the native expression.

->  Thank you for your comment. We revised the manuscript accordingly (Line 147-148).

  1. Line #161-166 “First, the direct visualization of the cyst wall and contents permits an accurate assessment of the extent of the lesion. Second, the endoscope permits an accurate assessment of the adequacy of the curettage, thus avoiding the need to perform multiple, blind, and aggressive passes with a curette, which can increase the risk of iatrogenic fracture. Third, the ability to completely evacuate the lesion should logically reduce the rate of recurrence. Lastly, it has advantages of a small incision, minimal blood loss, and limited dissection.”

The arrangement of some words in the text is similar to the arrangement of the following literature.

J Foot Ankle Surg. Jan-Feb 2010;49(1):93-7. doi: 10.1053/j.jfas.2009.08.005. Treatment of a unicameral bone cyst of calcaneus with endoscopic curettage and percutaneous filling with corticocancellous allograft
“First, direct visualization of the cyst wall and contents permits accurate assessment of the extent of the lesion. Second, the endoscope permits accurate assessment of the adequacy of the curettage, thus avoiding the need to perform multiple, blind, and aggressive passes with a curette, which can increase the risk of violation of the cortical shell and may prolong the procedure. Third, the ability to completely evacuate the lesion should logically reduce the rate of recurrence. In the patients depicted in this technique, the authors used endoscopic curettage because it provided optimal visualization with the advantage of a small incision, minimal blood loss, and a limited dissection.”

-> Thank you for your comment. We revised the manuscript into our own words (Line 158-165).

  1. Thank you for your hard work again.

-> Thank you for your generous consideration toward our work.

Reviewer 2 Report

The case report is well written.

The topic is interesting

Introduction is ok

Figures are exhaustive. However, please provide a post operative CT slice rather than a radiograph.

Table 1 please write "yes/no" instead of "0/1"

Discussion: please describe possible difficulties/complications when using medial access

Author Response

The case report is well written.

The topic is interesting

Introduction is ok

Figures are exhaustive. However, please provide a post operative CT slice rather than a radiograph.

-> Thank you for your comment. Unfortunately, patient refused to take a CT after surgery because of financial reason so there are no postoperative CT images available with us. I hope that you will generously understand such an inevitable situation.

Table 1 please write "yes/no" instead of "0/1"

-> Thank you. We revised the manuscript accordingly (Table 1).

Discussion: please describe possible difficulties/complications when using medial access

-> Thank you for your comment. We added the possible complication when using medial assess in the manuscript (Line 187-188)